# Identification of Candidate Genes for Red-Eyed (Albinism) Domestic Guppies Using Genomic and Transcriptomic Analyses

**DOI:** 10.3390/ijms25042175

**Published:** 2024-02-11

**Authors:** Ying Chang, Shenjun Wu, Junying Li, Haigang Bao, Changxin Wu

**Affiliations:** National Engineering Laboratory for Animal Breeding, Beijing Key Laboratory for Animal Genetic Improvement, College of Animal Science and Technology, China Agricultural University, Beijing 100193, China; 17545106088@163.com (Y.C.); 18855034676@163.com (S.W.); lijunying@cau.edu.cn (J.L.); chxwu@cau.edu.cn (C.W.)

**Keywords:** guppy, albinism, genomic analysis, transcriptomic analysis, *OCA2*

## Abstract

Guppies are small tropical fish with brightly colored bodies and variable tail shapes. There are two phenotypes of domestic guppy eye color: red and black. The wild type is black-eyed. The main object of this study was to identify candidate genes for the red-eyed phenotype in domestic guppies. We hope to provide molecular genetic information for the development of new domestic guppy strains. Additionally, the results also contribute to basic research concerning guppies. In this study, 121 domestic guppies were used for genomic analysis (GWAS), and 44 genes were identified. Furthermore, 21 domestic guppies were used for transcriptomic analysis, and 874 differentially expressed genes (DEGs) were identified, including 357 upregulated and 517 downregulated genes. Through GO and KEGG enrichment, we identified some important terms or pathways mainly related to melanin biosynthesis and ion transport. qRT-PCR was also performed to verify the differential expression levels of four important candidate genes (*TYR*, *OCA2*, *SLC45A2*, and *SLC24A5*) between red-eyed and black-eyed guppies. Based on the results of genomic and transcriptomic analyses, we propose that *OCA2* is the most important candidate gene for the red-eyed phenotype in guppies.

## 1. Introduction

Guppies (*Poecilia reticulata*) are small (adult size: 15–25 mm) ornamental fish with a wide range of colors and are very popular with consumers. They are native to freshwater streams in the West Indies, South American islands, Venezuela, Guyana, etc. [1,2,3]. In Singapore, about 30 guppy strains with different domesticated color patterns and/or tail-shape varieties are reared [4]. The male guppy is smaller and more colorful than the female of the same strain. The ages at maturity in the laboratory are two or three months [5]. Female guppies generally give birth once a month, and the generation interval of guppies in nature is around 100 days in complex fish communities and 180–200 days in fish communities living in smaller streams [6]. Guppies are a model fish in genetics and breeding for aquaculture studies [7,8,9] and are often used to investigate genetic drift, inbreeding, heterosis, adaptation, and the inheritance of quantitative traits [8,9]. Some researchers have focused on using guppies as an experimental model in evolution and sex selection [6,7]. In Trinidad, guppies were divided into two communities: high-predation and low-predation [10,11]. High-predation guppies were smaller and younger at sexual maturity and produced more offspring per litter [5], and female guppies of low predation showed stronger preferences for males with more orange/red coloration than females from high-predation environments [12,13].

Some domestic guppy strains appear to have a red-eyed phenotype, characteristic of albinism (Figure 1B). Human albinism is a group of heritable disorders associated with decreased or absent melanin in neural crest ectoderm-derived tissues, most notably in the skin, hair, and eyes [14]. Additionally, in human and zebrafish studies, it can be observed that eye color depends on the iris, and the iris has two layers: the IPE (iris pigment epithelium) and the iris stroma. The IPE originates from neuroectodermal and melanocytes, which are derived from the neural crest in the iris stroma [15,16]. The reason why the eyes appear red in this context is due to a lack of pigmentation, which causes the iris to become transparent and allows the color of the blood vessels in the eye to be reflected, resulting in an increased amount of light passing through the iris [17,18]. There are 12 kinds of albinism-associated disorders [14]: oculocutaneous albinism type 1 (OCA1), oculocutaneous albinism type 2 (OCA2), oculocutaneous albinism type 3 (OCA3), oculocutaneous albinism type 4 (OCA4), oculocutaneous albinism type 5 (OCA5), oculocutaneous albinism type 6 (OCA6), oculocutaneous albinism type 7 (OCA7), Hermansky–Pudlak syndrome (HPS), Chediak–Higashi syndrome (CHS), Angelman syndrome (AS), Prader–Willi syndrome (PWS), and Ocular albinism (OA1). To date, 22 genes are known to be related to albinism in animals [19]. 

There have been few reports of research on albinism in fish, and most of these involved medaka, zebrafish, and goldfish. The *TYR* mutation was shown to be the cause of albinism in medaka fish [20,21]. In total, seven loci of the *TYR* gene affecting melanin pigmentation have been described, and the corresponding mechanisms include deletion and transposable element insertion in the *TYR* gene [20,21,22,23,24,25]. Zebrafish constitute a model animal for studying albinism [26]. By constructing gene mutation models related to melanin production and transportation, the regulatory relationship between genes and albinism can be further unveiled [26,27]. At present, mutation models have been established for genes related to albinism [26], including the zebrafish models of *GPR143*(OA1) [28], *TYR*(OCA1a and b) [29,30], *OCA2/p*(OCA2) [30], *TYRP1*(OCA3) [31], *SLC45A2*(OCA4) [30,31,32,33], *SLC24A5*(OCA6) [34], *C10ORF11*(OCA7) [35], Hermansky–Pudlak Syndrome [36], Griscelli Syndrome [37,38], and other lysosome-related organelle disorders [39,40,41,42]. In goldfish studies, it has been found that two homologs of *OCA2* collectively control the albino phenotype [43,44]. These pattern studies of albino fish suggest that many genes, such as *GPR143*, *TYR*, *OCA2*, *TYRP1*, *SLC45A2*, *SLC24A5*, and *C10RRF11*, play important roles in the synthesis of fish eye pigments and that disturbances in their function lead to albinism and therefore may also be responsible for changes in fish eye color. 

Early reports indicated autosomal Mendelian inheritance of albinism [45]. However, the genes responsible for albinism in guppies have still not been identified. There is great potential for the development of ornamental fish for the consumer market, and the discovery of candidate genes that cause albinism in domestic guppies will provide new ideas for breeding new ornamental fish species in the future. As a model animal for many studies on evolution and fish genetics, the discovery of albino candidate genes will also fill a gap in basic research on guppies. In this study, we use genomic and transcriptomic analyses to identify candidate genes and propose that *OCA2* is the most important candidate gene for the red-eyed phenotype in domestic guppies.

## 2. Results

### 2.1. Overview of the Whole-Genome Sequencing Data

A summary of the whole-genome resequencing data is shown in Appendix A. The average clean base of each sample exceeded 10 G, and the Q20 value of each sample was above 94.72%. The average GC content was 39.76%. Through alignment with the reference genome of guppies (Guppy_female_1.0+MT), it was revealed that the average alignment rate of the samples was above 90%. After filtering, 121 guppies with 1,286,895 SNPs and 63 INDELs were retained for further analysis.

A summary of the transcriptome sequencing data is shown in Appendix A. The average clean base of each sample exceeded 7 G, and the Q20 value of each sample was above 95.41%. By aligning with the reference genome of guppies (Guppy_female_1.0+MT), it was determined that the average alignment rate of the samples was above 90%. These statistical data showed that the sequencing data were of good quality and could be used for subsequent analyses.

### 2.2. Genome-Wide Association Studies

Before performing genome-wide association studies (GWASs), we conducted principal component analysis (PCA) for all guppies. The results showed that the population distribution was relatively uniform, and there was no significant population stratification, as shown in Appendix A. The GWASs were performed based on 1,286,895 SNPs and INDELs of 121 guppies to find genomic regions associated with guppy eye color. We obtained an inflation factor of 0.981 based on the GWAS results, which showed that our population lacked significant population stratification. We found that 149 variants were significantly associated with guppy eye color (*p* < 3.885 × 10^−8^); a corresponding Manhattan plot is shown in Figure 2. The main association signal was in the region from position 13.91 Mb to 18.17 Mb on Chr4, containing 44 genes, namely, *NLGN1*, *NAALADL2*, *BARHL2*, *LRRC8DB*, *LRRC8C*, *KYAT3*, *LMO4B*, *SOX14*, *HDLBPA*, *BOKA*, *ATG4B*, *DTYMK*, *AGXTA*, *TTC39C*, *KIF1AA*, *KLHL30*, *SNED1*, *RYK*, *SLCO2A1*, *RAB6BA*, *CEP63*, *SAP130A*, *BCL6AA*, *LPP*, *TPRG1*, *TP63*, *ZBTB11*, *DPT*, *ATP1B1A*, *NME7*, *ZGC:172121*, *BIVM*, *SI:CH211-201H21.5*, *TMEM131*, *CNGA3A*, *UBE3A*, *CNGA3A*, *ATP10A*, *GABRB3*, *GABRA5*, *GABRG3*, *OCA2*, *HERC2*, and *ENSPREG00000018047.* A descriptive summary of the associated variants is shown in Table 1, and detailed information is provided in Appendix A.

### 2.3. RNA Sequencing Analysis and Quantitative Real-Time PCR

Before investigating the differences between red-eyed and black-eyed guppies at the transcriptional level, we designed a cross-experiment involving two strains of red guppies. The F1 results all showed red eyes. Next, we conducted differential expression analysis using the pipeline of HISTA2-StringTie-DESeq2, and the input file of the gene_count_matrix data (Appendix A) for DESeq2 was obtained using the following Python script: prepDE.py (https://ccb.jhu.edu/software/stringtie/index.shtml?t=manual, accessed on 4 July 2022). A total of 874 differentially expressed genes (DEGs) were identified (Appendix A), of which 357 were upregulated, and 517 were downregulated, as shown in Figure 3A. Some DEGs related to melanin expression between black-eyed and red-eyed guppies are shown in Figure 3B.

The functional annotation of DEGs was performed using DAVID under default settings, and the results are shown in Figure 3C,D as well as in Appendix A. In Figure 3C and Appendix A, we can see that the GO enrichment of many terms related to melanin biosynthesis processes, melanosome transmembrane transport, and vision are obviously enriched, such as ion transport (GO:0006811), melanosome membrane (GO:0033162), melanin biosynthetic process from tyrosine (GO:0006583), transmembrane transporter activity (GO:0022857), visual perception (GO:0007601), and photoreceptor activity (GO:0009881). In Figure 3D and Appendix A, we can see that the KEGG enrichment of some pathways mainly related to melanin biosynthesis and vision, such as tyrosine metabolism (hsa00350), melanogenesis (hsa04916), and retinol metabolism(hsa00830), are enriched, which is consistent with the above GO terms. 

Many DEGs, such as *OCA2*, *TYR*, *SLC45A2*, *SLC24A5*, *PMEL*, and *MC1R*, with functions related to melanin biosynthesis and metabolism [46,47,48,49,50,51], were mainly enriched in terms of the melanin biosynthetic process (GO:0042438), melanin biosynthetic process from tyrosine (GO:0006583), melanocyte differentiation (GO:0030318), melanosome membrane (GO:0033162), and melanosome (GO:0042470) in the present study. Among these DEGs, only *OCA2* was detected via GWAS (Table 1).

As the *TYR* gene is the key enzyme of the biosynthesis of melanin [46,52,53] and *OCA2*, *SLC452A*, and *SLC24A5* are involved in the ion transport of melanosomes [46,54,55,56], we performed qRT-PCR to verify the differences in the gene expressions levels of these genes between red-eyed and black-eyed guppies. In Figure 4, we can see that the expression level of *OCA2* was significantly lower in red-eyed guppies than in black-eyed guppies, while *TYR*, *SLC45A2*, and *SLC24A5* showed the opposite expression patterns to *OCA2* in red-eyed and black-eyed individuals. These findings are consistent with the results of the RNA sequencing analysis.

## 3. Discussion

Melanin is converted from tyrosine, which is derived from phenylalanine catalyzed by phenylalanine enzymes. Tyrosine enters the melanosome and is subsequently converted into dopaquinone by tyrosinase [57]. One portion of dopaquinone forms CysteinylDOPA in the presence of L-cysteine, which further transforms into pheomelanin [58]. Another portion first forms dopachrome, which then transforms into DHICA and DHI and, finally, transforms into eumelanin under the action of TYRP1, TYRP2, and TYR [52,59,60]. pH plays a critical role in melanogenesis [46,61]. Melanosome biogenesis involves the process of melanosome development and maturation, which comprises four distinct stages, each with a different internal pH [55,62]. Due to the proton-importing activity of the ubiquitously expressed vacuolar-type H(+)-ATPase, stage I and II melanosomes are acidic [46,54], and stage III–IV melanosomes exhibit a neutral pH [56]. Tyrosinase only exerts its optimal activity at a neutral pH, so the melanosomes must be neutral as they mature in order to promote optimal TYR activity [54,55,56]. It is highly possible that H^+^, Na^+^, Ca2^+^, Cu^+^, and Cl^−^ participate in melanosome stages III–IV [46,63]. The genes *OCA2*, *SLC45A2*, *SLC24A5*, and *TPC2* encode a variety of key proteins of ion transporters that are important in maintaining the pH of melanosomes [46,48,64,65,66]. 

OCA2 and SLC45A2 proteins are reported to be positive regulators of pH neutralization [64,67]. *OCA2* variants generate OCA2 in humans and pink-eyed dilution in mice [68]. OCA2 is an essential component of a melanosome-specific anion channel and mediates Cl^−^ conductance [46,64]. Tyrosinase activity is disrupted in albino zebrafish with a *SLC45A2* mutation but can be rescued via the re-injection of *SLC45A2* mRNA or treatment with bafilomycin [67]. *SLC45A2* is a member of the H^+^/sugar cotransporter family, making it the most promising H^+^ efflux transport protein [66]. “Golden” is a zebrafish mutant of *SLC24A5*. It has severely reduced pigmentation both in the skin and in the retinal pigment epithelium [34]. *SLC24A5* is thought to be a K^+^-dependent Na^+^/Ca2^+^ exchanger [48]. In addition, it may mediate Ca^+^ transport if expressed in melanosomes [69]. *TPC2* encodes the two-pore channel 2 protein. TPC2 is expressed in melanocytes and localizes to the melanosome-limiting membrane, and it likely regulates melanosomes’ pH and size by mediating Ca2^+^ levels [65,70]. Therefore, changes in the expression of these genes may affect the synthesis of melanin.

In the present study, we used GWAS to identify the candidate genes for the red-eye trait in guppies. As shown in Figure 2 and Table 1, the association signals are mainly located in the genomic region from 13.91 to 18.17 Mb on Chr 4. A total of 44 genes were identified, and further functional studies were performed. Two genes, *OCA2* and *HERC2*, caught our attention.

OCA2 (oculocutaneous albinism type II) is a type of oculocutaneous albinism, and its occurrence rate makes it the most common in the world (1:39,000) [14]. OCA2 is autosomal-recessive and is caused by mutations in the *OCA2* gene [50,71]. *OCA2*-encoding melanosomal transmembrane protein, also called p protein, is recognized as the homolog of pink-eyed dilution (p) in mice, constituting one of the first genes linked to hypopigmentation [68,72]. p protein is predicted to have 12 transmembrane domains and shares homology with anionic transport proteins [46,71]. Several researchers have proposed that OCA2 protein affects the pH level of melanosomes by mediating the conduction of Cl^−^, leading to the inactivation of tyrosinase (TYR) and subsequently preventing melanin synthesis [46,64,73]. Bellono et al. [64] used direct skin patch–clamp and eye melanosomes to identify novel chloride-selective anion conductance mediated by *OCA2* required for melanin production. *OCA2* expression increased pH, suggesting that chloride transport correlates with the pH of the melanosomes.

*HERC2*, also called *HECT* and RLD domain-containing E3 ubiquitin protein ligase 2, is a major contributor to human eye color variations [59,74,75]. Through a genome-wide association study, Eiberg et al. [76] found that the SNP in the intron 86 region of *HERC2* (rs12913832) has a higher association with blue eyes compared to SNPs on *OCA2* in Europeans [74,75]. The *HERC2* gene does not directly participate in melanin synthesis but plays an important role in regulating *OCA2* expression levels [75,76,77,78,79]. rs12913832 is located upstream of the *OCA2* promoter in a highly conserved sequence in the 86th intron of *HERC2*. In the case of the mutation of this SNP (rs12913832 A/G), the expression of the P protein encoded by *OCA2* decreases, effectively decreasing its effects on pigmentation [76]. 

In the present study, 874 DEGs were detected via differential expression analysis. Through GO and KEGG analyses, some important pathways or terms related to melanin synthesis, melanin metabolism, and ion transport were identified, including the melanin biosynthetic process (GO:0042438), tyrosine catabolic process (GO:0006572), melanin biosynthetic process from tyrosine (GO:0006583), melanocyte differentiation (GO:0030318), ion transport (GO:0006811), melanosome membrane (GO:0033162), melanosome (GO:0016020), transmembrane transporter activity (GO:0022857), organic cation transmembrane transporter activity (GO:0015101), tyrosine metabolism (hsa00350), and melanogenesis (hsa04916). In addition, in this study, we also enriched pathways such as the chloride channel complex (GO:0034707) and anion transmembrane transporter activity (GO:0008509). Based on the above enrichment results, we speculate that the abnormal ion transmembrane transport in the red-eyed guppy causes acidity in melanosomes and a complete lack of pigmentation in the eyes of guppies [46,64].

In the present study, several DEGs, such as *OCA2*, *TYR*, *SLC45A2*, and *SLC24A5*, were enriched in melanin biosynthetic processes (GO:0042438), transmembrane transport (GO:0055085), melanin biosynthetic process from tyrosine (GO:0006583), melanosome membrane (GO:0033162), and melanogenesis (hsa04916), and qRT-PCR was also performed to verify the differential expression levels of these genes between the red-eyed and black-eyed guppies. From Figure 3A,B and Figure 4, we observed a significant downregulation in the relative expression of *OCA2* and the expression of other genes involved in maintaining the pH of the melanosome that were significantly upregulated in red-eyed guppies. In a transcriptomic analysis of albino northern snakehead, some key melanogenesis genes showed significantly higher levels than in the WT northern snakehead [80]. The results were consistent with the gene expression trend of our identification of genes associated with melanin synthesis. It is likely that the feedback in terms of melanogenesis is due to the lack of melanin-based protection against UV radiation from light [80,81,82]. Integrated genomic and transcriptomic analyses were carried out (Table 1 and Appendix A and Figure 4), and we believe that it is possible that mutations in either the *OCA2* gene or the *HERC2* gene lead to the downregulation of *OCA2* expression in red-eyed guppies, resulting in melanosome acidity and tyrosinase being unable to catalyze the conversion of tyrosine to melanin.

Human OCA2 patients generally have varying degrees of congenital nystagmus, reduced vision acuity, refractive errors, and some degree of color vision impairment [14,18,19]. In this study, some terms in relation to vision were also brought to our attention: retinol metabolic process (GO:0042572), retinoic acid metabolic process (GO:0042573), visual perception (GO:0007601), and photoreceptor activity (GO:0009881). Therefore, we speculated that red-eyed guppies have similar eye symptoms in terms of ocular albinism.

## 4. Materials and Methods

### 4.1. Sampling and Sequencing

All domestic guppies used in this study were bred in the Laboratory of Animal Genetics and Breeding of China Agricultural University. The breeding conditions were as follows: 10 h of light, a 22–28 °C temperature range, and feeding twice a day. A total of 121 adult fish were sampled, including 62 red-eyed guppies and 59 black-eyed guppies. The red-eyed domestic guppies were from four strains, and the black-eyed guppies were from five strains, as shown in detail in Appendix A. One tail fin clip was collected from each guppy (adult guppies) and stored in ethanol. Genomic DNA was isolated from each sample using the TIANamp Genomic DNA Kit (Cat. #DP304, TIANGEN, Beijing, China) according to the manufacturer’s instructions. After quality control, the DNA samples were sent to a commercial company (BGI, Shenzhen, China) for next-generation sequencing. Whole-genome resequencing data with 150 bp paired-end reads were generated using a DNBseq platform. The resequencing depth of each sample was greater than 10× (Appendix A). Eye tissues were collected from 12 red-eyed and 9 black-eyed guppies of 6–8 months of age and immediately stored in liquid nitrogen. The red-eyed guppies were from two strains, and the black-eyed guppies were from three strains, as shown in detail in Appendix A. Total RNA was isolated using Trizol (Cat. #DP424, TIANGEN, Beijing, China) according to the Trizol protocol [83]. A Nanodrop 2000 was used to assess RNA quality and concentration. After quality control, RNA samples were sequenced using a DNBseq PE150 sequencing platform in BGI, too.

### 4.2. Genomic Variant Calling

After removing reads with low-quality bases containing adapters or poly-Ns from the raw data, the clean data were aligned with the reference genome (Guppy_female_1.0+MT.105) using bowtie2 software (v2.4.5) [84]; then, genome-wide single-nucleotide polymorphisms (SNPs) and small insertion–deletions (INDELs) were detected using the SAMtools (v1.9) “mpileup” module via the BCFtools (v1.9) “call” option [85].

### 4.3. Genome-Wide Association Studies

VCFtools (v0.1.17) was used to filter variants (SNPs and INDELs) according to the following criteria: minDP, 8; maxDP, 30; maf, 0.1; max-missing, 0.9; min-alleles, 2; max-alleles, 2; and thin, 300 [86]. Principal component analysis (PCA) was performed using GCTA software (v1.93.2) [87]. Genome-wide association studies were performed using GEMMA software (v0.98.1) with a mixed linear model (LMM) [88]. After Bonferroni correction, SNPs with a *p*-value of less than 0.05/total SNP number were considered to be significant SNPs. The Manhattan plot was created using the R package of qqman (v0.1.8). The annotation of significant SNPs was carried out with snpEff (v5.0c) based on the Guppy_female_1.0+MT assembly supported by Ensembl [89].

### 4.4. RNA Sequencing Analysis

To obtain high-quality reads, we performed quality-filtering procedures by removing reads aligned with the barcode adapter, i.e., reads with >1% unidentified nucleotides (N) and reads with >40% low-quality (Q ≤ 20) bases. After data quality control was performed, all RNA samples were aligned with the reference genome of Poecilia_reticulata (Guppy_female_1.0_MT.105) using Hisat 2 (v2.1.0) [90]. StringTie (v2.1.1) was used to assemble transcripts and estimate the gene expression levels [91]. Finally, differential expression analysis was performed using DEseq2 (v1.34.0) [92]. Genes with |log2(fold change)| > 1 and *p* < 0.05 were considered differentially expressed (DEGs). To further investigate the related functions of DEGs, we performed Gene Ontology (GO) and Kyoto Encyclopedia of Genes and Genomes (KEGG) enrichment analysis using DAVID (http://david.abcc.ncifcrf.gov/, accessed on 29 September 2022).

### 4.5. Quantitative Real-Time PCR

The eye tissues of 10 red-eyed and 10 black-eyed guppies at the age of about 6 months were collected. The total RNA of each sample of eye tissue was isolated as described above. About 500 ng RNA of each sample was used for cDNA synthesis using a reverse transcription kit (cat. #KR118, TIANGEN, Beijing, China). Gene *β-actin* was set as a reference control [93]. Primer sequences were designed using Primer3Plus (https://www.primer3plus.com/, accessed on 10 November 2022) and are shown in Table 2. Quantitative real-time PCR (qRT-PCR) was performed using qTOWER3 touch (Bio-Rad Laboratories, Inc., Hercules, CA, USA) with a 20 µL reaction system containing 10 µL of 2× SuperReal PreMix Plus (SYBR Green; Cat. #FP209, TIANGEN, Beijing, China), 0.6 µL of the forward primer (10 pmoL/µL), 0.6 µL of the reverse primer (10 pmoL/µL), 1 µL of cDNA template, and 7.8 µL of RNase free water. Each sample had two replicates. The thermal cycling process was as follows: 95 °C for 3 min and 40 cycles of amplification (95 °C for 5 s, Tm for 10 s, and 72 °C for 15 s). The relative expression quantification of each gene was calculated using the 2^−∆∆Ct^ method [94]. We used the t-test method to test the expression differences of the target genes using GraphPad Prism (v9.5.1; GraphPad Software, La Jolla, CA, USA).

## 5. Conclusions

In this study, we performed genomic and transcriptomic analyses to detect candidate genes for the red-eyed phenotype in domestic guppies. We found that a genomic region containing 44 genes between 13.91 Mb and 18.17 Mb of Chr 4 was statistically significantly associated with guppy eye color. After transcriptomic analysis, 874 DEGs were identified between the red-eyed and black-eyed guppies. Through GO and KEGG enrichment, we identified some important terms and pathways mainly related to melanin biosynthesis and ion transport. qRT-PCR was also performed to verify the differential expression levels of four important candidate genes between red-eyed and black-eyed guppies. Based on the results of the genomic and transcriptomic analyses, we propose that *OCA2* is the most important candidate gene for the red-eyed phenotype in guppies. Next, we will detect the polymorphism of the *OCA2* gene, establishing the association between mutation sites and phenotypes to further explain the downregulation of *OCA2* expression at the transcriptional level. Our study provides a preliminary explanation of the molecular mechanisms of some red-eyed domesticated guppies and identifies candidate genes that lead to ocular albinism, providing a theoretical basis for cultivating new strains of red-eyed domesticated guppies.

## Figures and Tables

**Figure 1 ijms-25-02175-f001:**
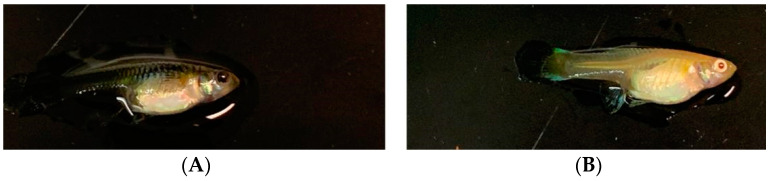
(**A**) black-eyed domestic guppy; (**B**) red-eyed domestic guppy.

**Figure 2 ijms-25-02175-f002:**
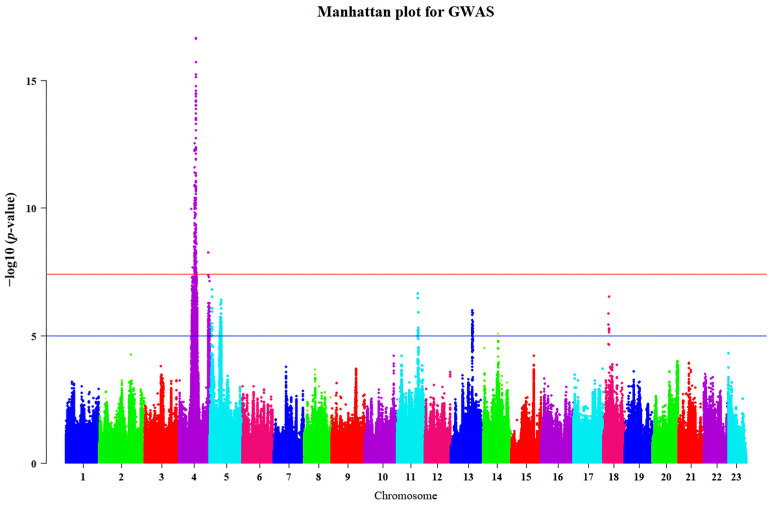
Manhattan plot for genome-wide association study of the eye color of guppies. The red line shows the *p*-value threshold of 3.885 × 10^−8^ (−log10(*p*-value) > 7.41) adjusted using the Bonferroni method with total SNP numbers after carrying out quality control [0.05/1,286,894]. The blue line is a suggestive line and the *p*-value is 1 × 10^−5^ (−log10(*p*-value) > 5).

**Figure 3 ijms-25-02175-f003:**
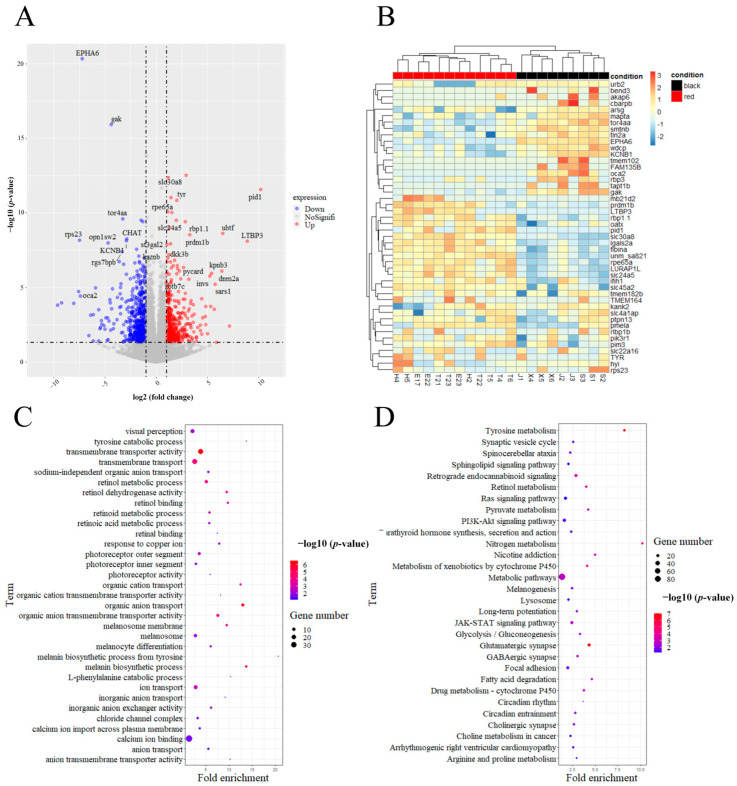
The figure shows a plot of DEG expression trends and the GO and KEGG enrichment results regarding the DEGs. (**A**) A volcano plot of the total expression of genes between red-eyed and black-eyed guppies. The blue dots represent genes downregulated in red-eyed compared to black-eyed guppies, and the red dots represent genes upregulated in red-eyed compared to black-eyed guppies. In red-eyed guppies, the *OCA2* gene is downregulated, and other related melanin biosynthesis genes, such as *TYR*, *SLC45A2*, *SLC24A5*, *MC1R*, etc., were upregulated relative to black-eyed guppies. (**B**) Heatmap of some DEGs between red-eyed and black-eyed guppies. (**C**) Some enriched GO terms of DEGs related to melanin biosynthetic, melanin metabolism, and vision. The small dots represent the number of genes enriched in the term. (**D**) The top 30 enriched KEGG pathways of DEGs. The small dots represent the number of genes enriched in the term.

**Figure 4 ijms-25-02175-f004:**
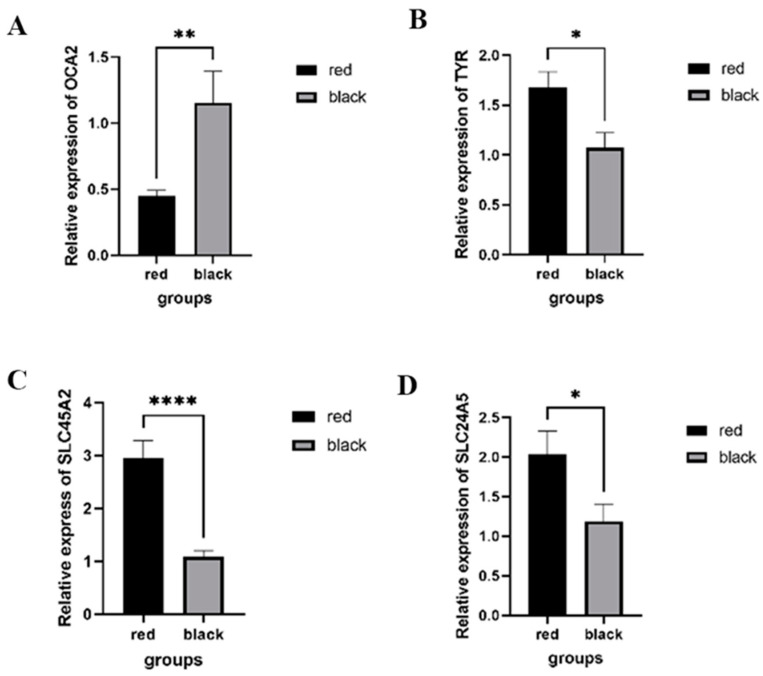
Relative expression levels of four DEGs between the red-eyed and the black-eyed guppies. (**A**) Relative expression of *OCA2*. The expression level was significantly lower in red-eyed guppies than in black-eyed guppies. (**B**) Relative expression of *TYR*. (**C**) Relative expression of *SLC45A2*. (**D**) Relative expression of *SLC24A5*. Data are shown as mean ± SD (*n* = 10). * represents *p* < 0.05; ** represents *p* < 0.01; **** represents *p* < 0.0001.

**Table 1 ijms-25-02175-t001:** A descriptive summary of significant variants associated with the eye color of guppies in GWAS.

Chr	Position (bp)	N_Sig ^a^	Lead Variant ^b^	*p* ^c^	Genomic Location	Corresponding Genes
4	13,919,679	1	13,919,679	2.13 × 10^−8^	intergenic	*NLGN1-NAALADL2*
4	15,579,097–157,04,885	2	155,79,097	3.34 × 10^−9^	intergenic	*BARHL2-LRRC8DB*
4	15,728,116	1	157,28,116	3.05 × 10^−8^	upstream	*LRRC8C*
4	15,735,128	1	15,735,128	3.23 × 10^−8^	intergenic	*LRRC8C-KYAT3*
4	15,815,011–15,929,479	28	15,908,422	2.87 × 10^−13^	intergenic	*KYAT3-LMO4B*
4	16,159,568	1	16,159,568	2.36 × 10^−8^	intergenic	*SOX14-HDLBPA*
4	16,365,953–16,388,992	5	16,379,136	1.89 × 10^−10^	upstream; intron; intergenic; downstream	*BOKA*
4	16,384,579–16,391,411	3	16,385,964	1.03 × 10^−10^	upstream; intron;	*ATG4B*
4	16,394,845–16,403,441	2	16,403,441	8.71 × 10^−10^	upstream; downstream	*DTYMK*
4	16,410,980–16,411,725	2	16,411,725	1.31 × 10^−8^	upstream	*AGXTA*
4	16,430,204	1	16,430,204	2.34 × 10^−8^	downstream	*TTC39C*
4	16,431,698–16,455,327	6	16,431,698	9.91 × 10^−11^	upstream; intron	*KIF1AA*
4	16,525,371–16,566,144	3	16,525,371	1.41 × 10^−9^	intergenic	*KLHL30* *-SNED1*
4	16,574,314–16,600,468	3	16,574,314	6.58 × 10^−9^	intron; CDS	*SNED1*
4	16,613,965–16,632,460	2	16,632,460	1.25 × 10^−9^	intron	*RYK*
4	16,675,710	1	16,675,710	3.32 × 10^−9^	intron	*SLCO2A1*
4	16,722,251	1	16,722,251	3.54 × 10^−8^	intron	*RAB6BA*
4	16,749,558–16,756,105	7	16,752,127	5.82 × 10^−11^	intron; downstream	*CEP63*
4	16,814,130	1	16,814,130	2.90 × 10^−8^	upstream	*SAP130A*
4	16,920,982–16,976,594	2	16,920,982	3.33 × 10^−8^	intergenic	*BCL6AA-LPP*
4	17,001,755	1	17,001,755	3.76 × 10^−8^	CDS	*LPP*
4	17,192,862	1	17,192,862	6.27 × 10^−9^	intron	*TPRG1*
4	17,243,325–17,250,077	2	17,250,077	9.00 × 10^−9^	intergenic	*TPRG1-TP63*
4	17,250,825–17,282,572	3	17,282,572	1.05 × 10^−10^	upstream; intron; downstream	*TP63*
4	17,276,094–17,292,722	5	17,289,603	6.20 × 10^−15^	upstream; downstream	*ZBTB11*
4	17,295,315–17,299,997	2	17,299,997	2.71 × 10^−8^	intergenic	*ZBTB11-DPT*
4	17,302,046–17,310,860	3	17,302,046	2.30 × 10^−17^	downstream; CDS; intron	*DPT*
4	17,317,551–17,328,988	8	17,320,328	1.89 × 10^−16^	intergenic	*DPT-ATP1B1A*
4	17,331,317–17,348,850	7	17,348,850	1.66 × 10^−15^	upstream; intron; downstream	*ATP1B1A*
4	17,356,626–17,372,435	8	17,369,146	2.26 × 10^−17^	downstream	*NME7*
4	17,366,454–17,378,171	3	17,378,171	7.54 × 10^−16^	downstream	*ZGC:172121*
4	17,381,538–17,383,905	3	17,381,538	4.39 × 10^−13^	intron; CDS	*BIVM*
4	17,398,644	1	17,398,644	1.57 × 10^−11^	downstream	*SI:CH211-201H21.5*
4	17,402,571–17,433,938	12	17,433,938	4.80 × 10^−13^	intergenic; upstream	*TMEM131*
4	17,446,682–17,446,995	2	17,446,682	2.20 × 10^−9^	intergenic	*TMEM131-CNGA3A*
4	17,458,944	1	17,458,944	3.36 × 10^−8^	downstream	*UBE3A*
4	17,461,407–17,462,566	2	17,462,566	3.36 × 10^−8^	downstream	*CNGA3A*
4	17,511,176–17,516,034	3	17,514,805	2.46 × 10^−8^	intron	*ATP10A*
4	17,548,494	1	17,548,494	4.67 × 10^−9^	intergenic	*ATP10A-GABRB3*
4	17,558,221	1	17,558,221	2.94 × 10^−8^	intron	*GABRB3*
4	17,594,093	1	17,594,093	3.29 × 10^−8^	intron	*GABRA5*
4	17,626,784–17,660,211	2	17,660,211	2.66 × 10^−9^	intron	*GABRG3*
4	17,703,618–17,709,085	2	17,703,618	1.26 × 10^−8^	intron	*OCA2*
4	17,770,644	1	17,770,644	1.25 × 10^−8^	CDS	*HERC2*
4	18,167,098	1	18,167,098	2.68 × 10^−9^	intron	*ENSPREG00000018047*

^a^ The number of significant variants, with *p* < 3.885 × 10^−8^; ^b^ The SNP with the smallest *p* at the corresponding position; ^c^ The *p* of lead variant.

**Table 2 ijms-25-02175-t002:** Primers used in qRT-PCR.

Primers	Forward Primer	Reverse Primer	References	Tm	Product Size (bp)
*β* *-actin*	gcttgtgcgggatatcatttg	gaatccggctttgcacatac	NM_001297475.1	60 °C	137
*OCA2*-2	cagactttcgggataacgcct	gagcactcctcctccgct	XM_008407632.2	60 °C	141
*TYR*-2	ctccatgtccaacgtccagg	catttgctcgtgggtagctg	XM_008425495.2	60 °C	131
*SLC45A2*-2	gagaggtctgcactaccacg	gtactcggagcccaacagac	XM_008423772.2	60 °C	115
*SLC24A5*-2	ttctcaggatgtggcaggag	tgctgattccaatgtccccc	XM_008404588.2	60 °C	110

## Data Availability

The DNA and RNA sequencing data for this study can be downloaded from the China National Gene Bank (Accession numbers: CNP0005066, CNP0005067).

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
