# Peer review of "Identification of Candidate Genes for Red-Eyed (Albinism) Domestic Guppies Using Genomic and Transcriptomic Analyses"

_ijms, 2024, doi:10.3390/ijms25042175_

Round 1

Reviewer 1 Report

Comments and Suggestions for Authors

The authors present a study of albinism (red-eyed phenotype) in guppies. As guppies have been domesticated a long time ago and many breeds have been created, there may not only be a single mutation that leads to albinism segregating in this fish's domestic population. Hence, the authors may not have found candidate genes for "the" red-eyed guppy (as suggested in the title), but only for "a" red-eyed guppy. And this already points to a general problem with the article: the authors should acquire more knowledge about their study system (the domesticated guppy, the cell biology of albinism and neural crest development) to guide their bioinformatic analyses and interpret their results.

A general strategy of bioinformaticians these days seems to be to use many methods that differ in their underlying assumptions and then take the intersection of the results. In this article, such a strategy is chosen for the centrally important results presented in Fig 2. A GWAS assumes a single panmictic population; an F_st analysis assumes two or more subdivided populations; and the Cross Population Extended Haplotype Homozogysity (XP-EHH) is used to detect selective sweeps in which the selected allele has approached or achieved fixation in one population but remains polymorphic in the population as a whole. Clearly the assumptions of the GWAS (a single panmictic population) and of the other two methods (subdivided populations) cannot hold simultaneously; at least one method must be wrongly applied to the current problem. It makes no sense to use the intersection of methods with mutually exclusive assumptions! Rather the authors need to decide depending on the population structure of the domestic guppy. This requires a review or a study of domestic guppy breeding. As I am not an expert on guppy breeding, I am not sure which assumptions fit a priori. But if I were to conduct such a study, determining the population structure would be the first step; locating genomic regions that influence the phenotype would follow only subsequently and depend on the results of the first step. In any case, the major candidate locus OCA2 seems reasonable, given functional information from other vertebrates.

mRNAseq-transcriptomics: It is quite likely that the red-eyed albino phenotype the authors study is caused by a mutation in a single "master" gene (which may well be OCA2). The mRNAseq analysis will also show many downstream changes in gene expression. These downstream genes may well be identical with the DEGs (differentially expressed genes) identified by the authors. Again, there is biology missing: little to no effort is spent to find out and explain if the DEGs are consistent with the observed phenotype. At first glance, it does not seem so to me as all genes necessary for melanin seem to be up-regulated in the red-eyed guppy. Are there mRNAseq studies available from other vertebrates (incl humans) that could serve as guidance?---As above, the authors need to invest more on the biology (as opposed to exclusively rely on bioinformatics).

The authors write about melanin in ectoderm-derived tissue; however melanophores are more specifically neural-crest derived. The importance of the neural crest for coat and skin color phenotypes has often been pointed out in relation to domestic animals (eg in pinto or paint horses). Indeed a theory for domestication is based on neural crest function (Wilkins AS, Wrangham RW, Fitch WT. The “domestication syndrome” in mammals: a unified explanation based on neural crest cell behavior and genetics. Genetics. 2014. Jul 1;197(3):795–808. doi:10.1534/genetics.114.165423). Hence, as pointed out above in another context: more biology, this time on neural crest development, would be needed.

I really suggest to add a (good) biologist as a co-author, who knows about domesticated guppy populations, can relate differentially expressed genes to the phenotype, and is informed on neural crest development. A reviewer can only point out deficiencies, but not fix them. This would add at least one more name to the author list, which seems to be quite heavy on bioinformaticians.

Comments on the Quality of English Language

Language: there are some grammatical errors and sometimes the wording could be improved. But generally I believe I understood the meaning. I attach a manuscript with some comments to this review.

Author Response

Dear Reviewer,

Thank you for your thoughtful comments. Please see the attachment.

Kind regards,

Ms Ying Chang

Reviewer 2 Report

Comments and Suggestions for Authors

Dear Editors,

Dear Authors,

In the reviewed manuscript the Authors conducted a comprehensive study to identify candidate genes associated with red-eyed phenotypes in guppies using genomic and transcriptomic analyses. The study successfully identified candidate genes associated with the red-eyed phenotype in guppies through a multi-faceted approach, i.e., the integration of GWAS, selection signature analysis and transcriptomic data. The findings suggest a significant role of genes involved in melanin biosynthesis, vision, and ion transport in albinisms that lies behind red eye color in guppies, particularly highlighting the importance of OCA2 gene.

The introduction is well-written, providing a solid foundation for the study. It effectively combines general information about guppies with specific details about the red-eye phenotype and its potential genetic basis. The accurate representation of existing knowledge and the clear proposal of the study's focus make it a strong and informative introduction. However, there are some discrepancies that needs should be clarified and some minor errors corrected:

1.     Explicitly state the research gap or need that the current study aims to address. Try to emphasize why understanding the genetic basis of the red-eye phenotype in guppies is significant?;

2.     Clearly state the research objectives and hypotheses that guide the study. This would provide a roadmap for readers that help them to understand the purpose and expected outcomes of your research;

3.     The sentence within the lines 64-67 looks like incomplete – the parentheses after the mutation models (e.g., OCA3 model) are not closed as well as lack of gene names are provided for Hermansky–Pudlak Syndrome, Griscelli Syndrome and other Lysosome-Related Organelle Disorders;

4.     I suggest to replace the word “variations” within the line 71 with “alterations or disturbences”;

5.     The sentences within the lines 73-75 are unclear. First you mention that in early reports the albino gene in guppies followed Mendelian inheritance and was located on autosomal chromosomes but in next sentence you say genes responsible for albino trait are still unknown. There is contradictory here. Please clarify the statement about the albino gene in guppies following Mendelian inheritance.

The Results chapter is well-structured, with a logical flow from sequencing data overview to detailed analyses. Figures and tables are appropriately used to enhance the clarity of the presented data. Statistical and analytical details are provided, contributing to the reproducibility of the study. The combination of genomics and transcriptomics analyses strengthens the study's approach to identifying candidate genes for the red-eyed guppy phenotype. Remarks and suggestions:

1.     In the figure 2D on Venn diagram is placed GEMMA term that is mentioned first time. It makes confusion. I suggest to replace it with GWAS;

2.     In table 1 the overlapping genes are not bolded as declared under it;

3.     Within the lines 142-144 there should be placed reference to figure 3A and 3B as it is not cited in the study;

4.     Within the lines 147-154 make clear distinction between the results of GO term and KEGG enrichments. Put correct reference to figure 3C and 3D;

5.     In the description of Figure 3 correct to capital letter within the line 164 “… guppy. the blue…”

The Discussion chapter of the reviewed study provides an in-depth analysis of the results, linking genetic findings to melanin synthesis and the red-eyed phenotype in guppies. The Discussion chapter is well-organized, providing a logical progression from genetic analyses to functional insights and potential phenotypic outcomes. My questions and suggestions:

1.     The information within the lines 197-200 require broader explanation. Please provide more information on functions of mentioned genes especially SLC45A2, SLC24A5 and TPC2;

2.     Information about HERC2 gene appears for the first time in the discussion chapter but the Authors did not mention about it in the abstract and in the results chapter;

3.     Word “initial” within the line 21 should be replaced with “first”?;

4.     Please extend the information within the lines 213-215 about the function of OCA2 gene. How this regulation takes place? By increasing or decreasing pH inside melanosome? According to your results how the under expression of OCA2 gene in red-eye fish may affect this regulatory mechanism?

5.     Line 218 “… SNP in intron 86 region…” You mean SNP located in 86th intron of this gene?

6.     The sentence within the lines 219-221 require further explanation. Why HERC2 gene particularly took your attention? I do not see it from now yet. How does this gene impact melanin synthesis?;

7.     Lines 230-233 - As you mentioned earlier - first stages of melanosome biogenesis takes place in acidic environment. Then, the pH increases to neutral at the end last two stages of melanosome biogenesis. Moreover, tyrosinase requires neutral pH for proper activity. So, maybe correct expression level of OCA2 gene is needed to make neutral pH inside the melanosome? According to your results, it looks like under expression of OCA2 gene contributes to disturbances in restring of neutral pH establishment inside melanosome and suppressing the activity of tyrosinase. Under such conditions, organisms may start to compensate it by upregulation of THY gene expression and other transporting genes (SLC-like) that brings inside melanosome required substrates for melanin synthesis? What is your opinion?;

8.     Line 234: Bellono et al. - the citation number is missing;

9.     Lines 235-237: For me there is missing information about the outcomes of OCA2 gene under or overexpression on melanosome pH and tyrosinase activity. Please provide more information if possible?;

10.  Lines 244-246: according to the figure 4 there is under expression of OCA2 gene and over expression of remaining genes. However, you wrote “…other genes that were significantly different between the red-eyed guppies and the black-eyed guppies.”

The Materials and Methods chapter is well-structured and provides a detailed account of the experimental procedures. The methods used are standard and widely accepted in the field, enhancing the reliability of the study. My minor suggestions:

1.     Line 260: “A total of 121 adult samples were collected …” replace with “A total of 121 adult fish were sampled …”;

2.     Line 261: “One tail fin was…: replace with “One tail fin clip was …”;

3.     Line 276: “… (v.2.4.5; …” close parenthesis “(v.2.4.5)”;

4.     Line 293: Place capital letter “… as positive signals. we …“ to “… as positive signals. We …“;

5.     Line 307: “… differentially expressed genes (DEGs).” replace with “… differentially expressed (DEGs).”

The Conclusions chapter provides a concise and clear summary of the study findings. Key findings are appropriately highlighted, including the proposed role of OCA2 gene in the red-eyed phenotype. Suggested improvements:

1.     According to the information within the lines 337-339 consider elaborating on the importance of investigating OCA2 polymorphism and how it contributes to understanding the phenotype;

2.     Include potential implications of the study findings for guppy genetics or other related research areas.

The abstract effectively communicates the main components of the study, including objectives, methods, and key findings. Clear language and organization make it accessible. Suggested improvements:

1.     Include a sentence indicating the potential broader implications or applications of the study findings.

The manuscript’s title is generally clear and descriptive of the study's focus. However, study is related to albinism, and I think title should include this aspect.

The manuscript appears to be well-organized, with a logical flow from the abstract through the different chapters. The research question is clearly addressed, and the results are interpreted in the context of existing knowledge. The combination of genomic and transcriptomic analyses enhances the comprehensiveness of the study. The writing is generally clear, and the use of figures and tables aids in conveying complex information. Overall, the manuscript appears to be a valuable contribution to the understanding of the genetic basis of the red-eyed phenotype in guppies. Therefore, I recommend to reconsider the article for possible publication in the IJMS periodical after major revision of especially discussion chapter.

Best regards,

Comments on the Quality of English Language

Dear Editors,

Dear Authors,

In the reviewed manuscript the Authors conducted a comprehensive study to identify candidate genes associated with red-eyed phenotypes in guppies using genomic and transcriptomic analyses. The study successfully identified candidate genes associated with the red-eyed phenotype in guppies through a multi-faceted approach, i.e., the integration of GWAS, selection signature analysis and transcriptomic data. The findings suggest a significant role of genes involved in melanin biosynthesis, vision, and ion transport in albinisms that lies behind red eye color in guppies, particularly highlighting the importance of OCA2 gene.

The introduction is well-written, providing a solid foundation for the study. It effectively combines general information about guppies with specific details about the red-eye phenotype and its potential genetic basis. The accurate representation of existing knowledge and the clear proposal of the study's focus make it a strong and informative introduction. However, there are some discrepancies that needs should be clarified and some minor errors corrected:

1.     Explicitly state the research gap or need that the current study aims to address. Try to emphasize why understanding the genetic basis of the red-eye phenotype in guppies is significant?;

2.     Clearly state the research objectives and hypotheses that guide the study. This would provide a roadmap for readers that help them to understand the purpose and expected outcomes of your research;

3.     The sentence within the lines 64-67 looks like incomplete – the parentheses after the mutation models (e.g., OCA3 model) are not closed as well as lack of gene names are provided for Hermansky–Pudlak Syndrome, Griscelli Syndrome and other Lysosome-Related Organelle Disorders;

4.     I suggest to replace the word “variations” within the line 71 with “alterations or disturbences”;

5.     The sentences within the lines 73-75 are unclear. First you mention that in early reports the albino gene in guppies followed Mendelian inheritance and was located on autosomal chromosomes but in next sentence you say genes responsible for albino trait are still unknown. There is contradictory here. Please clarify the statement about the albino gene in guppies following Mendelian inheritance.

The Results chapter is well-structured, with a logical flow from sequencing data overview to detailed analyses. Figures and tables are appropriately used to enhance the clarity of the presented data. Statistical and analytical details are provided, contributing to the reproducibility of the study. The combination of genomics and transcriptomics analyses strengthens the study's approach to identifying candidate genes for the red-eyed guppy phenotype. Remarks and suggestions:

1.     In the figure 2D on Venn diagram is placed GEMMA term that is mentioned first time. It makes confusion. I suggest to replace it with GWAS;

2.     In table 1 the overlapping genes are not bolded as declared under it;

3.     Within the lines 142-144 there should be placed reference to figure 3A and 3B as it is not cited in the study;

4.     Within the lines 147-154 make clear distinction between the results of GO term and KEGG enrichments. Put correct reference to figure 3C and 3D;

5.     In the description of Figure 3 correct to capital letter within the line 164 “… guppy. the blue…”

The Discussion chapter of the reviewed study provides an in-depth analysis of the results, linking genetic findings to melanin synthesis and the red-eyed phenotype in guppies. The Discussion chapter is well-organized, providing a logical progression from genetic analyses to functional insights and potential phenotypic outcomes. My questions and suggestions:

1.     The information within the lines 197-200 require broader explanation. Please provide more information on functions of mentioned genes especially SLC45A2, SLC24A5 and TPC2;

2.     Information about HERC2 gene appears for the first time in the discussion chapter but the Authors did not mention about it in the abstract and in the results chapter;

3.     Word “initial” within the line 21 should be replaced with “first”?;

4.     Please extend the information within the lines 213-215 about the function of OCA2 gene. How this regulation takes place? By increasing or decreasing pH inside melanosome? According to your results how the under expression of OCA2 gene in red-eye fish may affect this regulatory mechanism?

5.     Line 218 “… SNP in intron 86 region…” You mean SNP located in 86th intron of this gene?

6.     The sentence within the lines 219-221 require further explanation. Why HERC2 gene particularly took your attention? I do not see it from now yet. How does this gene impact melanin synthesis?;

7.     Lines 230-233 - As you mentioned earlier - first stages of melanosome biogenesis takes place in acidic environment. Then, the pH increases to neutral at the end last two stages of melanosome biogenesis. Moreover, tyrosinase requires neutral pH for proper activity. So, maybe correct expression level of OCA2 gene is needed to make neutral pH inside the melanosome? According to your results, it looks like under expression of OCA2 gene contributes to disturbances in restring of neutral pH establishment inside melanosome and suppressing the activity of tyrosinase. Under such conditions, organisms may start to compensate it by upregulation of THY gene expression and other transporting genes (SLC-like) that brings inside melanosome required substrates for melanin synthesis? What is your opinion?;

8.     Line 234: Bellono et al. - the citation number is missing;

9.     Lines 235-237: For me there is missing information about the outcomes of OCA2 gene under or overexpression on melanosome pH and tyrosinase activity. Please provide more information if possible?;

10.  Lines 244-246: according to the figure 4 there is under expression of OCA2 gene and over expression of remaining genes. However, you wrote “…other genes that were significantly different between the red-eyed guppies and the black-eyed guppies.”

The Materials and Methods chapter is well-structured and provides a detailed account of the experimental procedures. The methods used are standard and widely accepted in the field, enhancing the reliability of the study. My minor suggestions:

1.     Line 260: “A total of 121 adult samples were collected …” replace with “A total of 121 adult fish were sampled …”;

2.     Line 261: “One tail fin was…: replace with “One tail fin clip was …”;

3.     Line 276: “… (v.2.4.5; …” close parenthesis “(v.2.4.5)”;

4.     Line 293: Place capital letter “… as positive signals. we …“ to “… as positive signals. We …“;

5.     Line 307: “… differentially expressed genes (DEGs).” replace with “… differentially expressed (DEGs).”

The Conclusions chapter provides a concise and clear summary of the study findings. Key findings are appropriately highlighted, including the proposed role of OCA2 gene in the red-eyed phenotype. Suggested improvements:

1.     According to the information within the lines 337-339 consider elaborating on the importance of investigating OCA2 polymorphism and how it contributes to understanding the phenotype;

2.     Include potential implications of the study findings for guppy genetics or other related research areas.

The abstract effectively communicates the main components of the study, including objectives, methods, and key findings. Clear language and organization make it accessible. Suggested improvements:

1.     Include a sentence indicating the potential broader implications or applications of the study findings.

The manuscript’s title is generally clear and descriptive of the study's focus. However, study is related to albinism, and I think title should include this aspect.

The manuscript appears to be well-organized, with a logical flow from the abstract through the different chapters. The research question is clearly addressed, and the results are interpreted in the context of existing knowledge. The combination of genomic and transcriptomic analyses enhances the comprehensiveness of the study. The writing is generally clear, and the use of figures and tables aids in conveying complex information. Overall, the manuscript appears to be a valuable contribution to the understanding of the genetic basis of the red-eyed phenotype in guppies. Therefore, I recommend to reconsider the article for possible publication in the IJMS periodical after major revision of especially discussion chapter.

Best regards,

Author Response

Dear reviewer,

We have modified our manuscript according to your comments, thank you for your thoughtful comments.Please see the attachment.

Best regards,

Ms Ying Chang

Round 2

Reviewer 1 Report

Comments and Suggestions for Authors

As far as I could see, the authors did not provide detailed answers to my comments, but changed the manuscript according to them. So now the changed manuscript doubles as a rebuttal letter, it seems.

The manuscript now contains a lot of information that was missing before (on neural crest and color morph biology, biochemistry and cell biology, and guppy breeding), but still requires another round of revision.

Some minor comments I wrote directly into the manuscript, the major one here:

The authors now test for population structure and find out that there is none (l107ff). This means that the assumptions for the GWAS are met, but those for the other two methods XP-EHH and Fst are not. Hence those two analyses should be taken out. It seems enough to present the GWAS only, as the critical region with OAS2 and HERC2 seems also flagged as significant in the GWAS, where the region spans from 15.57 Mb to 17.77 Mb on Chr4. Asides from the scientific problem of showing results from methods, where the authors themselves show in the manuscript that their assumptions are not met, these analyses add 3 images, with an inconsistent color-code for the different chromosomes. As this is inappropriate and confusing, I strongly suggest leaving the Fst and the XP-EHH analyses and images out, at least of the main text. This also necessitates re-writing of the paragraph 235ff. Furthermore, I suggest to not use the term "populations" in line 339 when distinguishing between the two color morphs (rather use "morphs" or "variants"), as the authors' analysis shows that the two color morphs are not populations, in the sense that the term is usually used in population genetics. Note that GWAS, Fst and XP-EHH are population genetic methods and thus the population genetic context is appropriate.

Comments on the Quality of English Language

Generally I could understand easily, but many small suggestions are in the manuscript.

Author Response

Dear Reviewer,

Thanks for your comments. we accept your suggestion and have revised our manuscript. Please see attachement for more details.

Kind regards,

Ying Chang

Reviewer 2 Report

Comments and Suggestions for Authors

Dear Editors,

Dear Authors,

The manuscript entitled: “Identification of candidate genes for the red-eyed guppy using genomic and transcriptomic analyses” has been significantly improved. The Authors have regarded all my remarks and suggestions. I do not have any other remarks and the manuscript can be published after minor language corrections. For example: within line 286 sentence "The trends in these genes are consistent with our results" is repetition and should be deleted.

Good job!

Author Response

Dear Reviewer,

Thank you for your comments, we have changed line 262 which you mentioned. Thank you again for your help with this manuscript.

Kind regards,

Ying Chang